# Genome-Wide Identification and Evolutionary Analyses of *SrfA* Operon Genes in *Bacillus*

**DOI:** 10.3390/genes14020422

**Published:** 2023-02-07

**Authors:** Ying Xu, Jia-Yi Wu, Qing-Jie Liu, Jia-Yu Xue

**Affiliations:** 1State Key Laboratory of Enhanced Oil Recovery, PetroChina Research Institute of Petroleum Exploration & Development, Beijing 100083, China; 2College of Horticulture, Academy for Advanced Interdisciplinary Studies, Nanjing Agricultural University, Nanjing 210095, China

**Keywords:** *SrfA* operon, evolution, self-duplication, gene fusion, bacteria

## Abstract

A variety of secondary metabolites contributing to plant growth are synthesized by bacterial nonribosomal peptide synthases (NRPSs). Among them, the NRPS biosynthesis of surfactin is regulated by the *SrfA* operon. To explore the molecular mechanism for the diversity of surfactins produced by bacteria within the genus *Bacillus*, we performed a genome-wide identification study focused on three critical genes of the *SrfA* operon—*SrfAA*, *SrfAB* and *SrfAC*—from 999 *Bacillus* genomes (belonging to 47 species). Gene family clustering indicated the three genes can be divided into 66 orthologous groups (gene families), of which a majority comprised members of multiple genes (e.g., OG0000009 had members of all three *SrfAA*, *SrfAB* and *SrfAC* genes), indicating high sequence similarity among the three genes. Phylogenetic analyses also found that none of the three genes formed monophyletic groups, but were usually arranged in a mixed manner, suggesting the close evolutionary relationship among the three genes. Considering the module structure of the three genes, we propose that self-duplication, especially tandem duplications, might have contributed to the initial establishment of the entire *SrfA* operon, and further gene fusion and recombination as well as accumulated mutations might have continuously shaped the different functional roles of *SrfAA*, *SrfAB* and *SrfAC*. Overall, this study provides novel insight into metabolic gene clusters and operon evolution in bacteria.

## 1. Introduction

A nonribosomal peptide synthase (NRPS) is a multifunctional mega-enzyme that usually includes many modules and functions to assemble the peptide backbones of structurally diverse and biologically active natural products. It is well known that surfactin is one of the most important biosurfactants; it has the ability to efficiently decrease the surface tension of water and has good heat endurance and salt tolerance, meaning that it has great potential for use in enhanced oil recovery (EOR) [1,2,3]. The *SrfA* operon, consisting of four adjacent genes, *SrfAA*, *SrfAB*, *SrfAC* and *SrfAD*, is required for the production of surfactin [4]. Among these four genes, *SrfAA* and *SrfAB* are both composed of three functional modules, and *SrfAC* consists of only one module. Each module is comprised of an adenylation (A) domain, a thiolation (T) domain and a condensation (C) domain [5,6], and has the ability to incorporate a specific amino acid into a peptide backbone, so the order of the modules in assembly lines is always in accordance with the product peptide sequence. *SrfAD* contains a thioesterase (TE) domain, and is responsible for catalyzing the cyclization of peptide chains. Additionally, a few single-module NRPS-like enzymes that lack the important C domain have been reported in fungi and bacteria [7]. The A domain, C domain and T domain in the functional module have been associated with selective substrate activation, peptide bond formation and substrate shuttling in activation sites, respectively. They are vital for biosynthesis pathways, and the TE domain alone has been shown to be responsible for the release of compounds [8]. Thus, the *SrfAA, SrfAB* and *SrfAC* genes are considered as critical components in the *SrfA* operon.

*Bacillus* bacteria, known as a remarkable source of bioactive lipopeptides, are widely applied in the biosynthesis of a wealth of diverse compounds, including antibiotics, biosurfactants and antitumor agents via gene-cluster mining [9,10,11,12]. *Bacillus* bacteria produce a diversity of surfactins; however, it remains poorly understood how such a diversity is generated. In this study, we performed genome-wide identification of *SrfAA*, *SrfAB* and *SrfAC* genes from 999 *Bacillus* genomes, searched for conserved motifs in whole genes and conducted phylogenetic analyses to clarify the gene fusion of S*rfAA*, *SrfAB* and *SrfAC* genes in the *SrfA* operon, and proposed that self-duplication within the operon has been a driving force in the formation of the *SrfA* operon. Overall, our findings provide evidence that self-duplication could be an important mechanism in the formation of metabolic gene clusters and operon evolution in bacteria.

## 2. Materials and Methods

### 2.1. Data Source

A total of 999 *Bacillus* genomes (belonging to 47 species), including sequences and annotation files, were downloaded from the NCBI database (https://www.ncbi.nlm.nih.gov/, accessed on 7 June 2022). All data sources are shown in Appendix A.

### 2.2. Genome-Wide Identification of SrfAA, SrfAB and SrfAC Genes

SrfAA.hmm was built using hundreds of representative *SrfA* gene sequences from various bacteria downloaded from the NCBI database, using a hidden Markov model build (HMMbuild). SrfAB.hmm and SrfAC.hmm were built in the same way (Appendix A). Briefly, a two-step strategy was adopted to identify *SrfAA* genes. The first step was to perform an hmmsearch for protein sequences in each genome, with SrfAA.hmm as a query. Then, the remaining sequences were used to conduct a Blast search against representative *SrfAA* genes, using the amino acid sequences of every genome as a query for potential *SrfAA* genes. The threshold expectation value was set to 10^−4^ for the BLAST search. The same strategy was adopted for the identification of *SrfAB* and *SrfAC* genes.

### 2.3. Classification of Orthologous Genes and Phylogenetic Analysis

Orthologous genes encoded in 999 *Bacillus* genomes were classified using OrthoFinder (version 2.2.7) [13] with default parameters. *SrfAA*, *SrfAB* and *SrfAC* genes were selected from orthologous groups (OGs) containing *SrfAA*, *SrfAB* and *SrfAC* genes. Amino acid sequences of target genes were used for multiple alignments using ClustalW integrated in MEGA 7.0 with default parameter settings [14]. Overly divergent sequences were removed to prevent interference with the alignments and subsequent phylogenetic inference. Then, the resulting alignments were manually corrected using MEGA 7.0 for further improvement. Phylogenetic analyses were conducted using IQ-TREE and the maximum likelihood method [15]. The best-fit model was estimated using ModelFinder [16]. Branch support values were assessed using UFBoot2 tests [17]. The scale bar in the model indicates genetic distance.

### 2.4. Conserved Motif Analysis

The conserved protein motifs in the whole *SrfAA*, *SrfAB* and *SrfAC* genes were analyzed using Multiple Expectation Maximization for Motif Elicitation (MEME) and WebLogo with the default settings [18,19].

## 3. Results

### 3.1. Genome-Wide Identificaiton and Orthology Classification of SrfAA, SrfAB and SrfAC Genes

Altogether, 4680 *SrfAA* genes, 3522 *SrfAB* genes and 3558 *SrfAC* genes were identified from 999 *Bacillus* genomes using OrthoFinder. They were scattered across 66 orthologous groups (OGs) (Table 1), indicating that these genes can be classified into 66 evolutionary gene families and suggesting a great diversity within the *SrfA* operon. Different OGs contained varying numbers of genes, with two OGs (OG0000009 and OG0000006) accounting for nearly 50% (5864/11,760) of all identified genes; the 18 largest OGs contained 98.3% of the genes, and the remaining 48 OGs contained fewer than 15 genes each. *SrfAA*, *SrfAB* and *SrfAC* genes were distributed in 49, 45 and 35 OGs, respectively. Of the 66 OGs, 39 contained more than one gene type; 21 OGs contained *SrfAA*, *SrfAB* and *SrfAC* genes, 11 contained only *SrfAA* and *SrfAB* genes, 4 contained only *SrfAA* and *SrfAC* genes, and 3 contained only *SrfAB* and *SrfAC* genes (Appendix A). Such mixed compositions of genes suggest high sequence similarity between *SrfAA*, *SrfAB* and *SrfAC* genes, and implies potential intraspecies recombination within the same *SrfA* operon.

Among the OGs containing mixtures of gene members, the gene compositions were usually not balanced; sometimes *SrfAA* dominated the others, e.g., in OG0000009 and OG0000878, and sometimes *SrfAB* outnumbered the others, e.g., in OG0000006 and OG0000736. The largest family, OG0000009, had 1867 copies of *SrfAA* and 1269 copies of *SrfAB*, but had only 9 copies of *SrfAC*. Such discrepancy of gene numbers within OGs may suggest differential degrees of duplication events among species and/or specificity of different gene families.

### 3.2. Phylogenetic Analysis and Gene Duplications of SrfA Operon

Phylogenetic analyses were conducted on the 17 largest OGs containing members of all three gene families (OG0004815 was excluded because it had no *SrfAC* genes). In each of the constructed phylogenetic trees, it was observed that *SrfAA*, *SrfAB* and *SrfAC* genes often clustered with each other while seldom forming separate monophyletic branches for different gene types, suggesting a high identity of the sequences within the *SrfA* operon and close evolutionary relationships among different genes (Figure 1 and Appendix A). These results are in accordance with the above-mentioned high sequence similarity among the *SrfA* operon genes. All evidence supports a hypothesis that the complete operon is probably derived from self-duplication of only one gene, and further module fusion and recombination may have led to the formation of the complete operon.

Gene duplication events within the *SrfA* operon and the corresponding mechanisms were therefore analyzed. By randomly investigating 10 genomes, it was observed that a complete *SrfA* operon usually comprises multiple copies of *SrfAA*, *SrfAB* and *SrfAC* genes at a close distance (fewer than 50 bp) to each other, forming a gene cluster (Figure 2, Appendix A). These physically close genes are normally the results of tandem duplications, which have likely made the biggest contribution to the establishment of a complete operon; they do not result from segmental or ectopic duplications.

### 3.3. Conserved Motifs in SrfAA, SrfAB and SrfAC Genes

Although the three genes of the *SrfA* operon have very similar gene sequence, their functional roles are different and thus they can be expected to have evolved distinct important motifs that are critical to their differentiation of functions. To explore the critical motifs, we searched for conserved motifs in the amino acid sequences of *SrfAA*, *SrfAB* and *SrfAC* proteins. For each gene, the five most conserved motifs were identified and displayed using WebLogo. We found that some of these conserved motifs were shared by different genes, suggesting conserved functional roles in these regions. For instance, motif 1 was shared among all three genes, motif 4 was shared between *SrfAA* and *SrfAB*. There were conserved motifs specific to one gene, for instance, motif 2 showed absolutely different sequences among three genes, while motif 3 exhibited a homology among the three genes but had a low level of sequence similarity (Figure 3). Therefore, these divergent motifs may be derived from different regions or the results of different degrees of mutations among different genes, and should indicate functional divergence of the three genes.

## 4. Discussion

Microorganisms produce a large variety of products, the majority of which are derived via biosynthesis assembly lines known as NRPSs. The *SrfA* operon comprises a number of NRPS genes producing diverse metabolic products with a broad application, e.g., surfactin can be used to break the long carbon chains found in crude oil, degrading it into small molecules and thereby improving oil recovery [20]. *SrfAA*, *SrfAB* and *SrfAC* genes harbor entire modules (C-A-T) or didomains (e.g., T-A, T-C), the order and specificity of which determine the amino acid sequence in final peptide products [21,22]. Using a total of 999 *Bacillus* genomes, we identified all *SrfAA, SrfAB* and *SrfAC* genes and performed comparative and phylogenetic analyses to understand how these genes have evolved in the genus. The usual way in which prokaryotes acquire new genes is through horizontal gene transfer [3,23,24,25]. However, this study reported another potential mechanism—self-duplication, which does not require any novel genetic material from the environment or other organisms, and thus may be considered an alternative approach for genomic innovation.

Certainly, to equip a complete *SrfA* operon is a complicated process, and gene duplication is just the beginning. Duplicated gene copies need to fuse to form multimodule genes like *SrfAA* and *SrfAB*, which comprise three duplicated modules each, and different genes have to accumulate mutations to achieve functional divergence, allowing them to be functionally responsible for different metabolic products. Meanwhile, tandem duplications should result in high-density gene clusters, and the multiple gene copies of a cluster likely increase the expressional efficiency and improve the yield of final products. Recent studies have provided a body of evidence supporting the opinion that gene duplication serves as an important driving force in the evolution of operons in bacteria. For instance, gene duplication within the *mamAB* operon in *Alphaproteobacteria*, the *EPS* operon in Gram-negative bacteria, and PE and PPE gene duplication in the *ppe-pe* operon in *Mycobacterium* genus has resulted in functional diversification and increased complexity of biological pathways [26,27,28].

Among the studied *Bacillus* genomes, it is true that some species had a complete *SrfA* operon (the *Bacillus albus* operon comprised all four genes, *SrfAA*, *SrfAB*, *SrfAC* and *SrfAD*), and some had incomplete *SrfA* operons with certain genes missing (e.g., *Bacillus beveridgei*). However, a similar scenario can be observed within the same *Bacillus* species: certain stains/accessions of one *Bacillus* species had the complete *SrfA* operon (*Bacillus altitudinis* strain: Ba1449), but other strains/accessions of the same species possessed incomplete operons (*Bacillus altitudinis* strain: SCU11). This phenomenon reflects the highly active and complicated evolution of operons, which are subject to gene duplication, loss, rearrangement and horizontal gene transfer [29].

Moreover, a broad repertoire of synthetic compounds depends on module self-function and module–module interaction among the *SrfAA*, *SrfAB* and *SrfAC* genes. Whole-module deletions/insertions or individual domain changes in a module are likely to re-engineer the biosynthetic pathway, resulting in the generation of novel products. A previous study reported that to reorganize the molecular structure in gene clusters and to deliver new products, gene editing was applied to swap the conserved flavodoxin-like subdomain (FSD), which still preserved the overall structure of the A domain [30]. However, it is possible that deletion of domains was followed by module skipping [31]. To date, the engineering of biosynthesis pathways and production of novel peptides needs further exploration.

It is well known that the majority of the metabolic gene clusters in bacterial genomes are organized into operons and clusters, which probably benefits the co-regulation and co-expression of adjacent genes with related functions [32]. Additionally, gene order and structure in operons are vital to the products and synthetic efficiency of protein complexes [33]. Thus, complicated metabolic networks including regulatory checkpoints and cascades are dominant in bacterial metabolism. To sum up, our findings suggest another means by which genomic innovation can be achieved, and provide novel insight into bacterial genome evolution.

## Figures and Tables

**Figure 1 genes-14-00422-f001:**
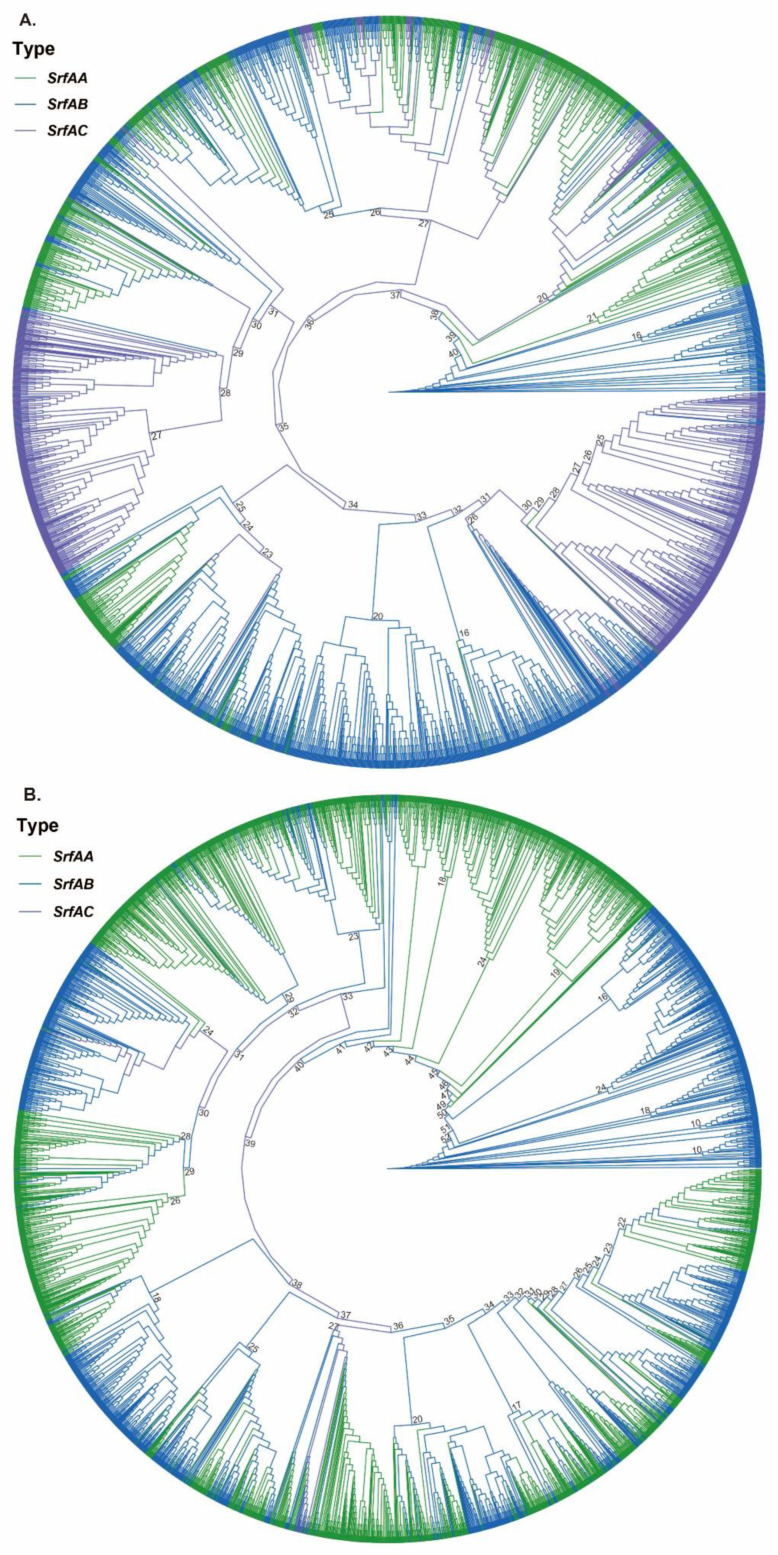
Phylogenetic trees of *SrfAA*, *SrfAB* and *SrfAC* genes in OG0000006 (**A**) and OG0000009 (**B**).

**Figure 2 genes-14-00422-f002:**
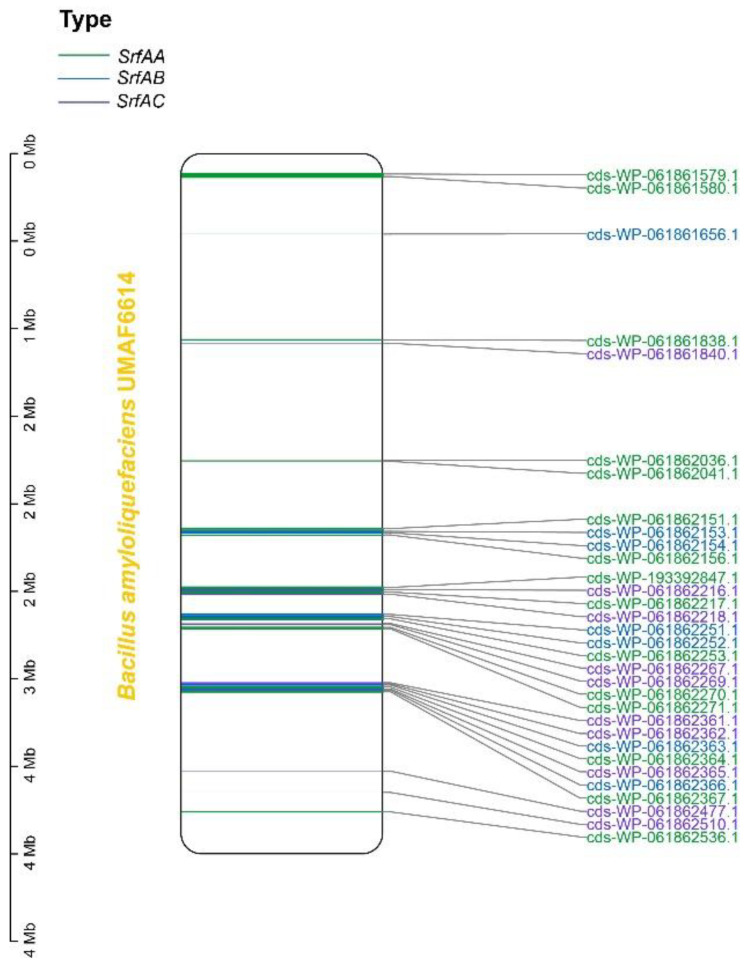
Distribution of *SrfAA*, *SrfAB* and *SrfAC* genes in *Bacillus amyloliquefacieus* UMAF6614 (GCF_001593785) genome.

**Figure 3 genes-14-00422-f003:**
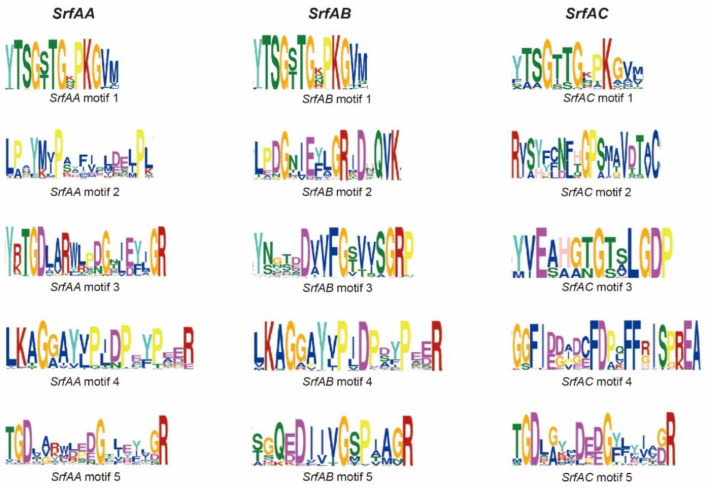
Conserved motifs of *SrfAA*, *SrfAB* and *SrfAC* proteins.

**Table 1 genes-14-00422-t001:** The number of *SrfAA*, *SrfAB* and *SrfAC* genes in 66 orthogroups.

	*SrfAA*	*SrfAB*	*SrfAC*	Total
OG0000003	0	1	0	1
OG0000006	854	1137	748	2739
OG0000009	1867	1249	9	3125
OG0000201	1	0	0	1
OG0000249	47	320	564	931
OG0000642	19	1	422	442
OG0000675	78	59	315	452
OG0000736	69	177	95	341
OG0000753	142	9	195	346
OG0000878	562	6	2	570
OG0000898	256	5	87	348
OG0001033	1	3	228	232
OG0001616	34	9	267	310
OG0002144	59	251	28	338
OG0002176	49	6	291	346
OG0002286	197	151	42	390
OG0002670	1	0	0	1
OG0003280	7	5	181	193
OG0003557	124	11	33	168
OG0004686	94	2	13	109
OG0004752	0	1	0	1
OG0004815	141	40	0	181
OG0010209	7	2	0	9
OG0010252	0	9	0	9
OG0010478	0	5	0	5
OG0010878	8	2	4	14
OG0010995	4	4	1	9
OG0011013	3	6	2	11
OG0011500	1	0	0	1
OG0011540	4	3	5	12
OG0011966	1	0	0	1
OG0012128	4	7	0	11
OG0012871	0	2	0	2
OG0013096	1	0	2	3
OG0013144	6	3	0	9
OG0013585	1	0	0	1
OG0013645	1	7	0	8
OG0013763	3	0	0	3
OG0013843	2	0	6	8
OG0013958	0	2	1	3
OG0014074	0	4	2	6
OG0014114	4	0	1	5
OG0014447	0	3	0	3
OG0014449	4	0	0	4
OG0014601	1	0	3	4
OG0014611	0	0	2	2
OG0014625	0	0	3	3
OG0014743	3	3	0	6
OG0014864	1	5	0	6
OG0015393	2	1	0	3
OG0015566	3	0	0	3
OG0017220	0	3	1	4
OG0017439	0	0	2	2
OG0017660	2	2	0	4
OG0018022	3	0	0	3
OG0018042	2	0	0	2
OG0018115	3	0	0	3
OG0018719	1	1	0	2
OG0019324	1	1	0	2
OG0019998	2	0	0	2
OG0020386	0	1	0	1
OG0020490	0	0	1	1
OG0020635	0	2	0	2
OG0022852	0	1	0	1
OG0036272	0	0	1	1
OG0036370	0	0	1	1

## Data Availability

The datasets generated for this study can be found in the National Center for Biotechnology Information (NCBI) (https://www.ncbi.nlm.nih.gov/, accessed on 7 June 2022).

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
