# Peer review of "Genome-Wide Identification and Evolutionary Analyses of SrfA Operon Genes in Bacillus"

_genes, 2023, doi:10.3390/genes14020422_

Round 1
Reviewer 1 Report
Manuscript ID : GENES-2073330
Title: Genome-wide identification and evolutionary analyses of SrfA operon genes in Bacillus
The authors present the in silico analysis of the SfrA operon in a large collection of Bacillus assembly. They analysed the sequence to build orthologous groups and to compare them through phylogenetic analysis. Due to sequence similarity, they concluded on the role of duplication in the arising and spread of SfrA operon throughout the Bacillus genus.
Comments: Major revision
The subject fulfils the journal topic and brings interesting view on the SfrA gene family among Bacillus. Nevertheless, some points need to be addressed before publication.
The material & method section 2.4 stated that PFAM and SMART analysis have been performed on structural motifs found in the tree SfrA gene family. No results of such experiment are shown in the proposed manuscript. Such experiment would bring interesting data on the functional differences between the SfrAA, SfrAB and SfrAC. It could be interesting to discuss along with the variety of surfactins.
If not done, the statement has to be removed.
When several assemblies are available for many species, it could be mentioned the total number of analysed species.
The quality of figure 1 and 3 could be improved., especially figure 3 that is difficult to read. It could be enlarged
The title of figure 2 does not show the assembly number it arises from.
There is a need of homogeneity in the text mention of figures, supplemental figures and tables.
l.96 : less/more
l.177 : reformulate
Reviewer 2 Report
The manuscript describes the identification and evolutionary analyses of the SrfA operon genes genome-wide within the genus Bacillus. The analyses were well-done. The orthologous gene analysis clustered the three genes, SrfAA, SrfAB and SrfAC, into 66 orthologous groups. The authors also showed that using phylogenetic analysis the three genes did not form monophyletic groups suggesting a close relationship among the three genes. Given the high sequence similarity, the author proposed the involvement of self-duplication self-duplication (e.g. tandem duplications) in the initial creation of the SrfA operon within the genus Bacillus. Did the analysis show any species dependent association? If so the authors should consider to discuss that in the “Discussion” section. Also, the “Discussion” section needs additional information of previous publications for and against the suggestion that the initial establishment of such operon could be due to gene duplication.
Minor comments:
-Line 14: “… Bacillus bacteria”, also extensively in the text could be “…. Bacillus spp. …” or “… bacteria within the genus Bacillus …”
Line 14: what made SrfAA, SrfAB and SrfAC critical genes? Please, add a sentence in the “Introduction” to explain.
Lines 41-43: Something is not grammatically correct in this sentence. Please, rephrase. For example “ Additionally, a few single ……enzymes that lack ……………..”
Lines 52-54: Not sure the authors provided direct evidence to definitely say self duplication is an important means ….." The authors should consider adding “…could be …” in the sentence.
Line 68: why was the threshold expectation value 10-4 selected?
Line 71: Please, provide a reference for OrthoFinder version 2.2.7
Lines 75-77: Please rephrase for clarity especially “… that too divergent …”
Lines 79-80: Should the reader expect the branch support values assessed using UFBoot2 be displayed on the trees of Figure 1?
Lines 122: could “…looked…” be “… looking …”
Round 2
Reviewer 1 Report
Manuscript ID : GENES-2073330
Title: Genome-wide identification and evolutionary analyses of SrfA operon genes in Bacillus
The authors present the in silico analysis of the SfrA operon in a large collection of Bacillus assembly. They analysed the sequence to build orthologous groups and to compare them through phylogenetic analysis. Due to sequence similarity, they concluded on the role of duplication in the arising and spread of SfrA operon throughout the Bacillus genus.
Comments: Acceptation
The subject fulfils the journal topic and brings interesting view on the SfrA gene family among Bacillus.
The authors take into account the reviewer remarks.